# The Problem of Compressive Strength in Direction Perpendicular to the Grains on Example of Tests of the Load-Bearing Capacity of the Continuously Supported Timber-Frame Sill Plates

**DOI:** 10.3390/ma13051160

**Published:** 2020-03-05

**Authors:** Janusz Brol, Jan Kubica, Marek Węglorz

**Affiliations:** Department of Structural Engineering, Silesian University of Technology, Akademicka 5, 44-100 Gliwice, Poland; janusz.brol@polsl.pl (J.B.); jan.kubica@polsl.pl (J.K.)

**Keywords:** compression perpendicular to the grain, effective contact area, timber, spruce, Eurocode 5

## Abstract

This paper presents a discussion of the problem of compressive strength in a direction perpendicular to the grains based on test results of the joints made by timber posts and sill plate. These tests accompanied a larger series of full-scale tests of timber frame walls. The test elements were made of solid softwood (spruce). The wood moisture was low, which corresponds to the real working conditions of these elements in the walls of a building (low humidity is typical for dry wood in the built-in wall of a real building). In the tests, the compression strength of timber perpendicular to the grain was exceeded in the sill plate in the area in contact with the posts. Shortly before reaching the state of failure, large displacements in the sill plate were measured on the contact surface with the post, and the grains in the sill plates were cut off at the edge of the post. The full-scale test results showed an overestimation of the load-bearing capacity in compression perpendicular to the grain when calculated on the basis of EN 1995-1-1+A1:2008 (Eurocode 5), and, therefore, the need to modify the current approach for determining it.

## 1. Introduction

Light timber frame walls are recognized as a very efficient and reliable wooden structure. In particular, their stiffness increases when they are joined with wood-based panels [1,2,3]. Nevertheless, many of structural aspects still require further recognition. In designing timber frame structures, one of the most important problems is the proper verification of the stress state at the joints between the timber posts and the sill plate. The present authors discuss this problem based on the experimental results of tests carried out at the Department of Structural Engineering, Silesian University of Technology, Poland. The main aim of these tests is to evaluate the compression stresses and deformations perpendicular to the grain at the sill plates of the timber frame. Selected results from testing full-scale timber frame wall members are presented. The test results were compared with the current approach given in EN 1995-1-1+A1:2008 (Eurocode 5) [4]. The research accompanying the tests was contracted by the Wood Core House Company, Jaworzno, Poland.

### 1.1. State of the Art

The design model adopted in the Eurocode 5 is currently based on the tests made by Madsen et al. [5] and the later modification of this model by Blass and Görlacher [6]. According to this model, the load-bearing capacity of wood results from the compressive strength perpendicular to the grain *f_c_*_,90,*k*_ the effective contact area *A_ef_*, and the factor *k*_*c*,90_, which takes into account the load configuration, the possibility of splitting, and the degree of compressive deformation.

According to Eurocode 5, the effective contact area *A_ef_* is determined by the effective contact length with the grain; namely, the contact length *l* which shall be increased by a maximum of 30 mm at each side of the contact area.

The load-bearing capacity of wood is modified by the *k_c,_*_90_ parameter that accounts for influencing factors such as the moisture content, wood species, and load case. The value of *k_c,_*_90_ should be taken as 1.0, but might be increased gradually to 1.75 with respect to the load case. For members on continuous supports, Figure 1 shows that for *l*_1_ ≥ 2*h* the value of *k_c,_*_90_ should be taken as 1.25 for solid softwood timber and 1.5 for glued-laminated softwood timber. Leijten et al. [7] have questioned these discontinuities in the assumption of the *k_c,_*_90_ coefficient based on empirical models and proposed different formulae to derive its value on the basis of the physical model by Van der Put [8].

The compressive strength of timber in the direction perpendicular to the grain (CPG), *f_c_*_,90,*k*_ is one of the crucial parameters affecting the load-bearing capacity of wood. The mean modulus elasticity of wood across the grain is smaller by a factor of about 30 than when parallel to the axis of the tree (longitudinal) [9]. In addition, elongated cells of wood are stiffer and stronger when loaded along the axis of the cell rather than when loaded across it (Bodig and Jayne [10], Dinwoodie [11]). The problem is the lack of a unified approach to determining the standard CPG strength. For example, as Leijten [12] states, in the Scandinavian countries, the standard characteristic bearing strength is 2–3 times higher than the stress at the proportional limit determined by tests. This bearing strength results from the assumption of a very high short-term CPG strength value equal to 6.5 MPa (in relation to 2.5 MPa in Eurocode 5) for spruce wood. This value is considered questionable and very conservative. Moreover, the CPG strength properties vary in radial and tangential directions [13,14,15]. The CPG strength is also associated with wood density. This is particularly evident when comparing sound wood to insect-deteriorated wood [16].

A common standard test procedure for the CPG strength is prescribed in EN 408+A1:2012 [17]. In this standard, the CPG ultimate load capacity, *F_c,_*_90_*_,max_* is defined as the intersection of a line (1) parallel to the linear part of the load, *F*–displacement, Δ*h* curve; a line (2) that is off-set by 1% of the standardized specimen depth, *h* (Figure 2). The force corresponding to the upper limit of the linear segment of the load, *F*–displacement, Δ*h* curve is known as the proportional limit, *F_c,_*_90_*_,prop_*. This approach comes from former prescriptions of the fiber stress at the proportional limit, or the stress which causes a 1% deformation, identified by Kolmann and Côté [18].

This standard procedure was used by Leijten [12] for testing spruce wood. He compared three different computational models to determine the CPG strength for deformations from 1% to 10% of the thickness of the element. He proved that the Eurocode 5 model substantially overestimates the CPG strength at 1% deformation, whereas it provides a good prediction at 10% deformation. In contrast, Leijten emphasized the accuracy of the physical model based on yield slip-line theory by Van der Put [8]. This physical model was recognized as flexible in accounting for the differences in beam height and load configuration, and predicts the compressive strength at 3% and 10% deformation accurately, unlike the empirical models.

### 1.2. Objectives

The main idea for this article came from the full-scale tests of the timber wall frame members made of European spruce wood (Picea abies). During these tests, the load-bearing capacity of the wood was exceeded in the sill plate at the contact area with the post.

The general objective of this study is to examine the load-bearing capacity in the compression perpendicular to the grain of solid softwood timber in the full-scale structural element, and to compare it with the requirements of the current standard, Eurocode 5.

Whereas it is common to test specimen using metal-on-wood compression [14,16,19,20], the wood-on-wood compression in the full-scale timber frame wall was investigated. There is evidence that the metal-on-wood compression inaccurately reflects the typical wood-on-wood compression often present in structural applications [21].

Two measurement techniques were used: a traditional measuring system using linear variable transducers (LVDT) and digital image correlation (DIC). The usefulness of the DIC method in testing the structural elements has already been proved [22,23,24].

## 2. Materials and Methods

### 2.1. Research Materials and the Test Members Geometry

Investigations were carried out on the individual members of a timber frame wall: the sill plates loaded by the posts. These elements were connected by spiral nails. The tested timber frames were made of a solid C24 class structural timber according to Eurocode 5. The timber plates and posts were joined longitudinally using glued finger-joints according to EN 15497:2014 [25]. The joints were located randomly alongside the members, but were not present at the tested connections. Considering that these tests accompanied a larger series of full-scale tests of timber frame walls, the geometry was specified by the timber frame system. This system is going to be used in residential buildings. The timber frame was sheathed on one side with 15 mm thick oriented strand boards (OSB) or gypsum fiberboards, based on the specifications of the tested timber frame wall system. The OSB and gypsum fiberboards were connected to the timber frame using 1.8 mm × 45 mm staples. The staple spacing was 150 mm for the end posts and sill plates, and 300 mm for the internal posts. The cooperation of the timber frame members with the OSB or gypsum fiberboards was negligible. The test used 60 mm × 160 mm sill plates supported continuously on 80 mm × 160 mm timber plates, as well as 60 mm × 160 mm top plates.

European spruce wood (Picea abies) with a low humidity was used in this research. The wood moisture varied in all series in the 10%–12% range. This humidity range is typical for dry wood built into the wall of a real building.

### 2.2. Research Methodology

The tests were carried out at a laboratory in the Silesian University of Technology in Gliwice, at about (22 ± 2) °C and about (40 ± 5)% relative humidity.

A schematic of the bottom part of the test stand is shown in Figure 3.

Two tests series were used:(a)In Series 1, the timber frame consisted of two 40 mm × 160 mm side posts and one 80 mm × 160 mm middle post at the clear distance length of 545 mm. In this series, 16 timber frame wall members were tested. Compression perpendicular to the grains was evaluated only for the side joints, which means that 32 joints marked as “L” and “R” were considered (Figure 3a);(b)In Series 2, the timber frame consisted of two 40 mm × 160 mm side posts at the clear distance length of 545 mm. In this series, 16 timber frame wall members were tested, which means that 32 joints marked as “L” and “R” were considered (Figure 3b).

The test scheme reflected the real working conditions of the timber frame wall in a real building, including its joints with the sill plate. In the tests, compression perpendicular to the grain was applied to the posts in the bottom sill plates and the top plates. Due to omitting the additional bending and/or buckling effects, only the bottom sill plate was analyzed. In the tests, the sill plate was supported continuously at the bottom of the test stand.

During the tests, a vertical, uniformly distributed load was applied to the top plate, as shown in Figure 2. The load in the full-scale tests of the timber frame wall was applied vertically using a steel beam and a hydraulic cylinder (stroke extension up to 100 mm and loading force up to about 1400 kN) mounted to the steel frame beam. For the loading-force registration, a previously calibrated electro-resistance dynamometer mounted at the top, under the hydraulic cylinder, was used. A fragment of the test stand is shown in Figure 4.

The load was applied in a static manner, at a speed of approximately 2 kN/min. The testing time was approximately (300 ± 120) s. In the tests, the failure modes and deformations of the sill plates that were loaded perpendicular to the grain by vertical posts were analyzed. The tests were carried out until sill plate failure occurred.

The timber frame wall members used during the test are shown in Figure 5. For the selected timber frame wall members, the side surfaces were painted for use with the ARAMIS DIC system.

Inductive gauges (LVDTs) were used to measure deformations in several parts of the full-scale timber frame walls. Only external bottom LVDTs were used to measure local deformations in the sill plates, due to compression stresses made by the posts, as shown in Figure 6.

Additionally, displacements were measured using an ARAMIS DIC system, but only for some test samples.

## 3. Results

A typical graph of the force (*F*)–displacement relationship of the sill plate at the connection with the post is shown in Figure 7. As a result of the stresses acting across the fibers, three phases can be distinguished in the work of a sill plate: (a) the initial, elastic wood deformation (Line 1 in Figure 7) and subsequent hardening curve; (b) the linear force–displacement relationship (Line 2 in Figure 7); (c) the nonlinear and ductile force–displacement relationship leading to failure.

The value of the maximum compressive load perpendicular to the grain, *F_c,_*_90_*_,max_*, should be determined according to EN 408+A1:2012 [17]. To determine *F_c,_*_90_*_,max_* it is suggested to draw the intersection of the *F*–Δ*h* plot with a 1% off-set of the standardized specimen depth (Line 3 in Figure 7). This means that in the case of a sill plate depth of 60 mm the off-set is equal to 0.6 mm.

Nevertheless, in testing the timber frame wall member, the static scheme was much more complex, when compared to the requirements of EN 408+A1:2012 [17], for determining the maximum compressive load perpendicular to the grain, *F_c,_*_90_*_,max_*. Taking into account the initial hardening of the test specimens, the proportional limit value of the compression force perpendicular to the grain, *F_c,_*_90_*_,prop_*, for each joint was considered for further analysis, instead of the maximum compressive load perpendicular to the grain, *F_c,_*_90_*_,max_*.

The obtained experimental results are given in Table 1.

The statistical analysis of the test results is given in Table 2.

The *F*–displacement, Δ*h* relationships of the sill plate at the connection with the post are shown in Figure 8.

The regression curves shown in Figure 8 and derived from the test results in Series 1 and Series 2 were used to determine the mean proportional limit value of the compression force perpendicular to the grain, *F_c,_*_90_*_,prop,mean_*. The value of *F_c,_*_90_*_,prop,mean_* was determined at the start of the nonlinear and ductile force–displacement relationship.

During the tests, clear dents were created in the sill plates, and these dents were limited to the contact area between the post and the sill plate. In these wooden dents, the wood fibers in the sill plates were cut off at the contact edges of the posts and the sill plates, see Figure 9.

For the displacement measurements for some selected timber frame wall members carried out using ARAMIS DIC, the measurement points (“facet points”) were defined in the DIC evaluation software and a viewer for measurement results—GOM Correlate program [26] approximately every 10 mm along the edge of the sill plate.

An image and the displacement diagram for the individual measuring points on the edge of the sill plate at the connection with the post over the duration of the test are shown in Figure 10 and Figure 11, respectively.

As the ARAMIS DIC system was used only for a few timber frame wall members, it was not necessary to present quantitative results. Nevertheless, the exemplary plots are presented to show the development of displacements at the contact length of the sill plate and the post over the duration of the test.

The results are presented for the examples of “joint 2.4L” and “joint 2.4R”. The displacements of the edge of the sill plate are shown in the elastic phase and just before failure. The reference force *F* = 26.78 kN was marked in these plots. This force value defines the point when the significant differences in displacements for the individual measuring points at the joint and outside the joint began to occur. As a result of these differences in displacements, at failure the fibers in the sill plate were cut off at the edge of the post.

From the statistical analysis of the test results presented in Table 2 we obtained the values of the characteristic compression force perpendicular to the grain in the tests as *F_c,_*_90_*_,prop,k,test_* equal to 21.10 kN in Series 1 and 24.34 kN in Series 2, respectively. These values are compared with the characteristic compression force perpendicular to the grain, *F_c,_*_90_*_,k_*, derived from Eurocode 5 formulae.

The timber compressive strength perpendicular to the grain, *f_c,_*_90_*_,k_*, was determined for timber class C24:*f_c,_*_90_*_,k_* = 2.50 MPa(1)

The effective contact area, *A**_ef_*, assuming a one-sided increase of the contact length parallel to the grain, *l*, by 30 mm and 160 mm width of the sill plate, *b* (Figure 12):*A**_ef_* = (*l* + 30)*b* = (40 + 30)∙160 = 11,200 [mm^2^](2)

Hence, the characteristic compression force perpendicular to the grain, *F_c,_*_90_*_,k_*, according to Eurocode 5:*F_c,_*_90_*_,k_* = *k_c,_*_90_ ∙ *f_c,_*_90_*_,k_* ∙ *A**_ef_* = 1.25 ∙ 2.5 ∙ 11,200 = 35,000 [N] = 35.0 [kN](3)

Assuming the recommended value of the factor, *k_c,_*_90_, which takes into account the load configuration, the possibility of splitting, and the degree of compressive deformation:*k_c,_*_90_ = 1.25(4)

In general, the effective contact length, *l*, in Equation (2) is considered as the contact length in compression increased by 30 mm from each side of the contact length. This rule applies to all wood species considered by Eurocode 5. It means that the effective contact length for *l* = 40 mm wide side posts in the tests shall be assumed as *l**_ef_* = 40 + 30 = 70 mm. According to the test results for the spruce wood, the effective contact length was in fact smaller, as presented in Figure 10 and Figure 11.

With respect to the test results, the Eurocode 5 overestimates the load-bearing capacity in the compression strength perpendicular to the grain. The characteristic proportional limit value of the compression force perpendicular to the grain in the tests was lower than the characteristic compression force perpendicular to the grain according to Eurocode 5 by about 40% in Series 1 and 30% in Series 2 (Table 3). Thus, the present authors confirmed, in the full-scale tests of the timber frame wall members, Leitjen’s results [12] of the spruce wood tests, which were completed using the standardised specimens according to EN 408+A1:2012.

## 4. Conclusions

During the full-scale tests of the timber frame wall members, it was significant that the compression stresses perpendicular to the grain at the joints of the sill plates and posts were high, and that they were decisive in the failure of the whole timber frame. Further analyses and the spruce wood that was previously tested demonstrates that the current standard EN 1995-1-1+A1:2008 (Eurocode 5) [4] overestimates the load-bearing capacity in the compression perpendicular to the grain. In many design situations, this can be dangerous, because it decreases the safety level of the structural elements.

The characteristic proportional limit value of the compression force perpendicular to the grain in the tests was lower than the characteristic compression force perpendicular to the grain according to Eurocode 5 by about 40% in Series 1 and 30% in Series 2. This agrees with the earlier investigations of the CPG strength of spruce wood made by Leijten [12]. Thus, it seems that there is a need to modify the current Eurocode 5 approach for determining the load-bearing capacity in compression perpendicular to the grain, with respect to the type of wood.

## Figures and Tables

**Figure 1 materials-13-01160-f001:**
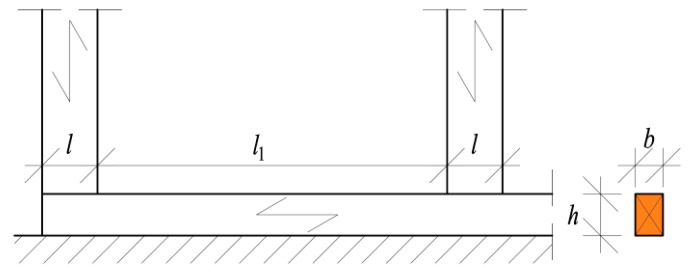
Loading conditions in case with continuous supports according to EN 1995-1-1+A1:2008 [4].

**Figure 2 materials-13-01160-f002:**
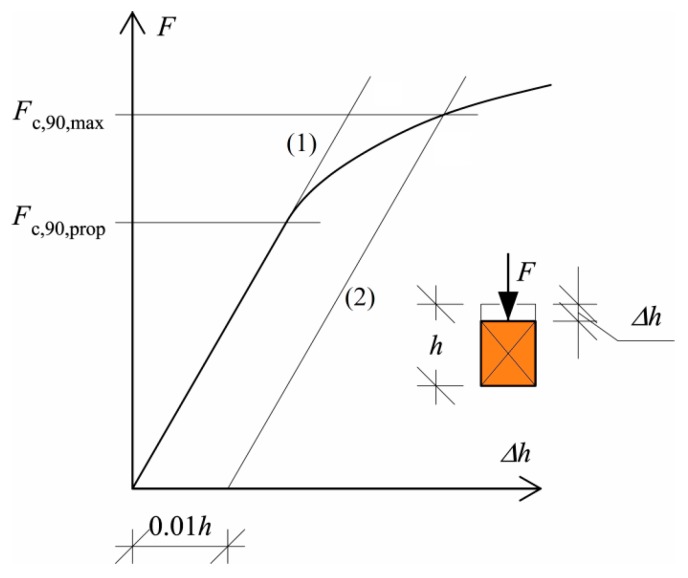
Determination of *F_c,_*_90_*_,prop_* according to EN408+A1:2012 [17].

**Figure 3 materials-13-01160-f003:**
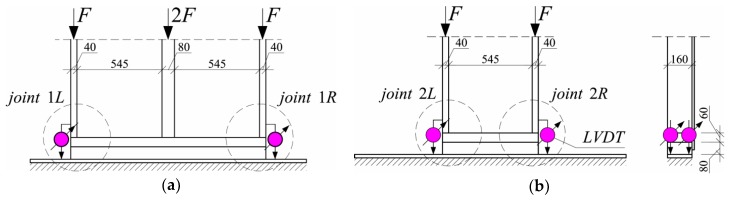
Schematic of the bottom part of the test stand: (**a**) Series 1, (**b**) Series 2, (dimensions in mm).

**Figure 4 materials-13-01160-f004:**
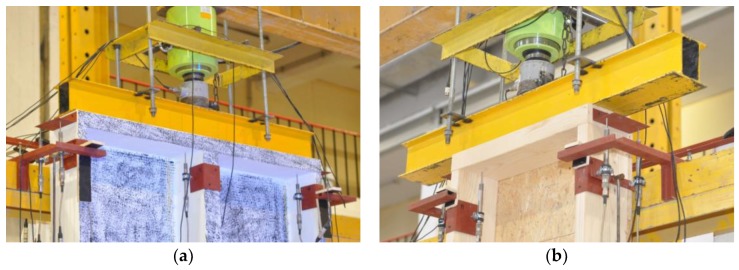
The top part of the test stand (load distribution): (**a**) Series 1, (**b**) Series 2 (photographs by Brol).

**Figure 5 materials-13-01160-f005:**
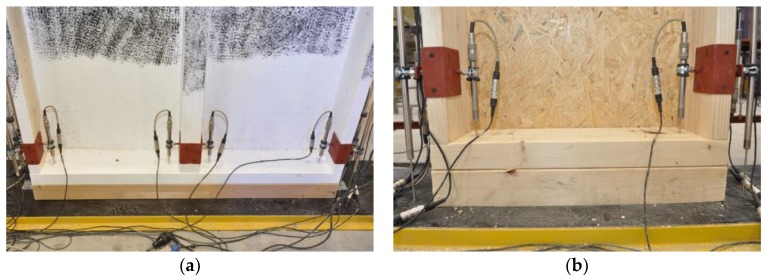
The bottom part of the test stand, support conditions: (**a**) Series 1 (painting is due to the ARAMIS DIC), (**b**) Series 2 (photographs by Brol).

**Figure 6 materials-13-01160-f006:**
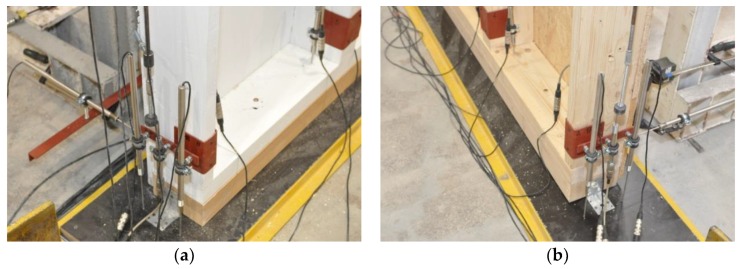
The arrangement of the external LVDTs: (**a**) joint L, (**b**) joint R (photographs by Węglorz).

**Figure 7 materials-13-01160-f007:**
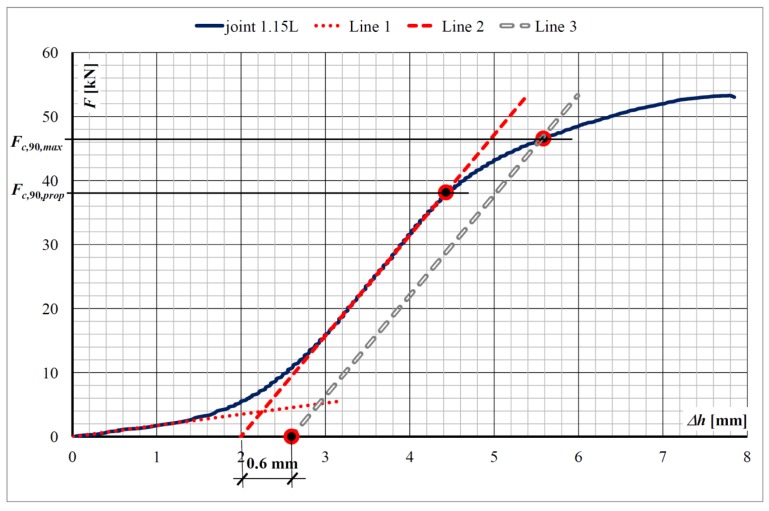
Exemplary force (*F*)–displacement Δ*h* plot of the sill plate at the connection with the post (joint 1.15L).

**Figure 8 materials-13-01160-f008:**
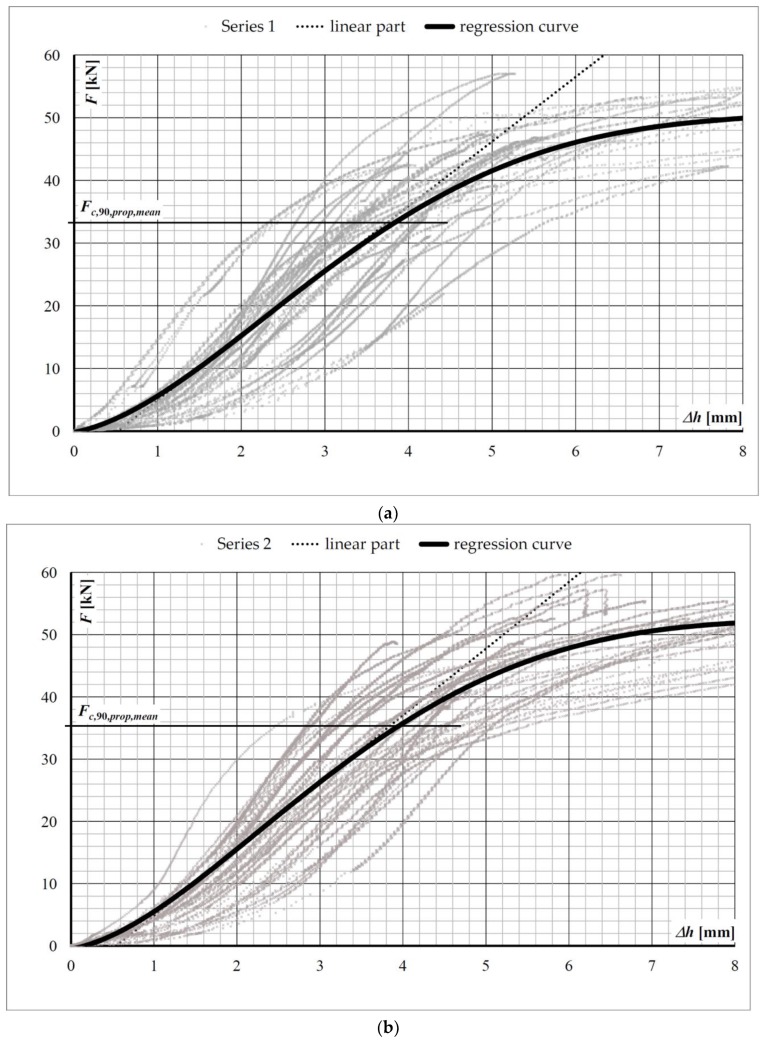
The *F*–displacement, Δ*h* plot of the sill plate at the connection with the post: (**a**) Series 1 results and regression curve, (**b**) Series 2 results and regression curve. The mean compression force perpendicular to the grain, *F_c,_*_90*,prop,mean*_, according to Table 2.

**Figure 9 materials-13-01160-f009:**
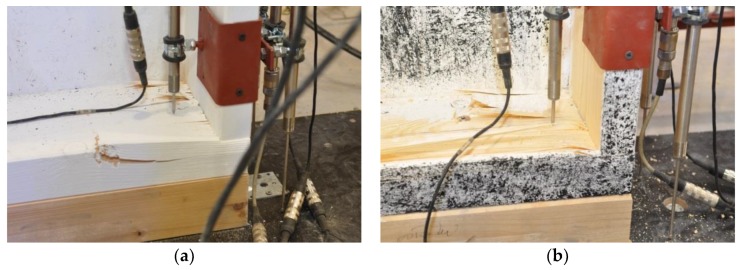
The failure mode of the joints of the timber frame wall members: (**a**) Series 1, (**b**) Series 2 (photographs by Brol).

**Figure 10 materials-13-01160-f010:**
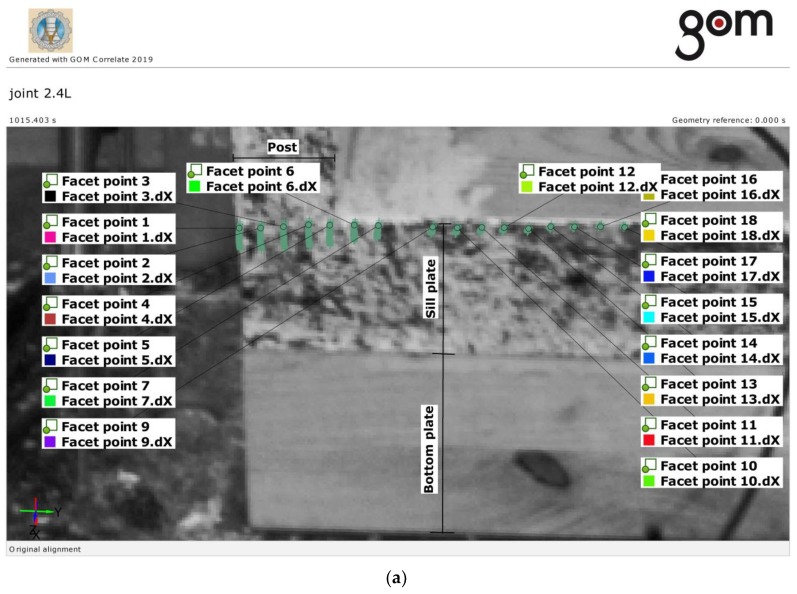
Exemplary displacements dX at the edge of the sill plate (joint 2.4L), over the duration of the test (ARAMIS DIC): (**a**) reference points and their trajectories (green), (**b**) deformation plot. The force *F* = 26.78 kN at total DIC running time (1015.403 s) is marked by the vertical line (red).

**Figure 11 materials-13-01160-f011:**
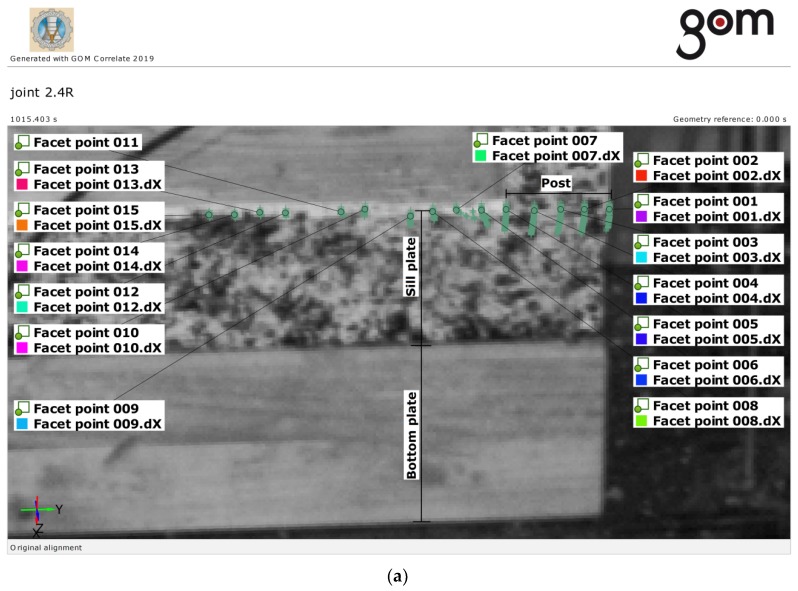
Exemplary displacements dX at the edge of the sill plate (joint 2.4R), over the duration of the test (ARAMIS DIC): (**a**) reference points and their trajectories (green), (**b**) deformation plot. The force *F* = 26.78 kN at total DIC running time (1015.403 s) is marked by the vertical line (red).

**Figure 12 materials-13-01160-f012:**
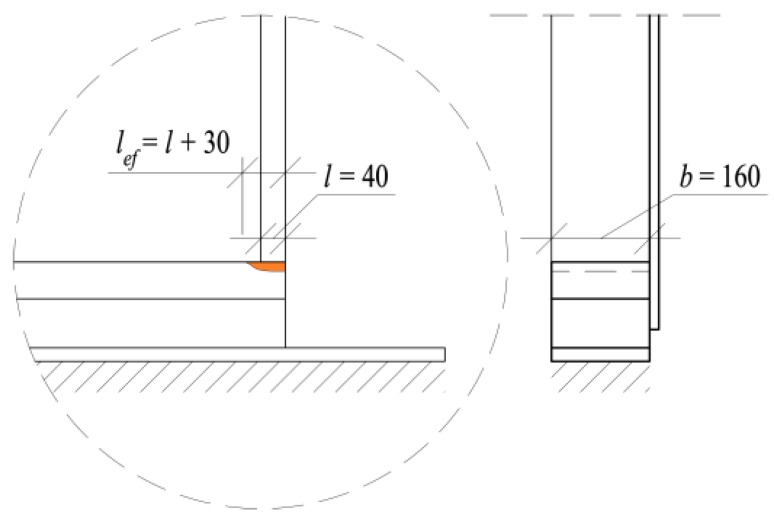
Trail specimen dimensions specified for the calculation of the effective contact area, *A_ef_* according to EN 1995-1-1+A1:2008 [4] (dimensions in mm).

**Table 1 materials-13-01160-t001:** The test results of the post–sill plate joints.

Timber Frame Wall Member Number	Proportional Limit *F_c,_*_90*,prop*_ [kN]			
	Series 1		Series 2	
	Joint “L”	Joint “R”	Joint “L”	Joint “R”
1	42.24	46.01	26.07	35.08
2	38.21	38.93	28.35	29.04
3	26.84	32.91	26.33	27.84
4	44.39	44.39	26.53	26.66
5	26.18	28.12	32.49	31.48
6	37.34	36.85	31.07	28.04
7	30.20	34.71	38.23	38.40
8	26.93	25.22	41.65	43.70
9	27.99	26.02	34.16	39.55
10	23.92	47.29	42.38	41.28
11	25.95	27.04	35.06	37.06
12	28.20	38.83	35.93	36.44
13	38.69	36.34	48.89	46.92
14	26.93	32.32	32.11	31.45
15	38.15	28.36	41.68	33.98
16	27.81	28.02	40.80	41.21

**Table 2 materials-13-01160-t002:** The statistical analysis of the test results.

Statistical Value	Series 1	Series 2
Mean compression force perpendicular to the grain, *F_c,_*_90*,prop,mean*_ [kN]	33.17	35.31
Maximum compression force perpendicular to the grain, *F_c,_*_90*,prop,max*_ [kN]	47.29	48.89
Minimum compression force perpendicular to the grain, *F_c,_*_90*,prop,min*_ [kN]	23.92	26.07
Standard deviation, *s* [kN]	6.98	6.34
The *T_s_*-value vs. the critical *T**_α_* -critical value check, *T_s_*(*F_c,_*_90*,prop,min*_) = |*F_c,_*_90*,prop,min*_−*F_c,_*_90*,prop,mean*_|/*s* < *T**_α_* = 2.75	1.33	1.46
*T_s_*(*F_c,_*_90*,prop,max*_) = |*F_c,_*_90*,prop,max*_−*F_c,_*_90*,prop,mean*_|/*s* < *T**_α_* = 2.75	2.02	2.14
Coefficient of variation *ν, ν* = *s*/*F_c,_*_90*,prop,mean*_	0.21	0.18
Characteristic compression force perpendicular to the grain, *F*_c,90,*prop,k,test*_, *F_c,_*_90*,prop,k,test*_ = *F_c,_*_90*,prop,mean*_−*k_n_* × *s* [kN], where the minimum-variance unbiased estimator, *k_n_* = 1.73	21.10	24.34

**Table 3 materials-13-01160-t003:** A comparison of the test results and Eurocode 5.

Feature	*F**_c_*_,90,*prop,k,test*_[kN]		*F**_c_*_,90,*k*_[kN]	*F**_c_*_,90,*prop,k,test*_/*F**_c_*_,90,*k*_ Ratio	
	Series 1	Series 2	Eurocode 5	Series 1	Series 2
Characteristic compression force perpendicular to the grain	21.10	24.34	35.0	0.60	0.70

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
