# Peer review of "The Problem of Compressive Strength in Direction Perpendicular to the Grains on Example of Tests of the Load-Bearing Capacity of the Continuously Supported Timber-Frame Sill Plates"

_materials, 2020, doi:10.3390/ma13051160_

Round 1

Reviewer 1 Report

This paper presents compression tests on the full-scale tests of the timber frame wall members.Overall, the paper is designed properly. The test is simple but obtained results is significant and meaningful. The study not only confirms the previous Leitjen’s results, but also provide a good evidence for the need of modification of the Eurocode 5's equation for determining load bearing capacity in CPG with respect to the kind of wood. I recommend this paper for publication after addressing below minor points.

  1. The section 1.2 is not smooth. The authors should amphasize a bit more on the contribution of this study against the previous works.
  2. The GOM correlate program should be cited in the text and included in the reference list.
  3. Provide the specs of the glued finger-joints in the timber frames used in the test
  4. Is there any future works related to this topic?

Author Response

Firstly, the authors are grateful for all valuable comments received from the reviewer which allow improving our article!

Our answers to the reviewer’s comments and remarks are given below, in a point-by-point manner.  

Ad. 1. The section 1.2 is not smooth. The authors should amphasize a bit more on the contribution of this study against the previous works.

Thank you for this valuable remark!

The references about the previous works have been added.

In particular, e.g. such works have been cited: Malesza et al. 2019, Branco et al. 2017, Sartori et al. 2013, Madsen et al. 2000, Blass & Görlacher 2004, Leijten 2010, Van der Put 2008, Bodig & Jayne 1982, Dinwoodie 2000, Leijten 2016, Požgaj et al. 1993, Kretschmann 2008, Tabarasa & Chui 2001, Verbist et al. 2020, Kollmann & Côté 1968, Leijten & Jorissen 2010, Kathem et al. 2014, Basta et al. 2011, Stoilov et al. 2019, Speranzini et al. 2017, Johansson et al. 2015.

DOI numbers of some referenced articles are as follows:

https://doi.org/10.3846/jcem.2019.7738

https://doi.org/10.3390/buildings7030063

https://doi.org/10.1016/j.engstruct.2013.08.039

https://doi.org/10.3390/buildings10010014

https://doi.org/10.1515/HF.2011.104

https://doi.org/10.3390/ma10070770

https://doi.org/10.1007/s00107-014-0851-3

Ad. 2. The GOM correlate program should be cited in the text and included in the reference list.

Thank you for your comment!

The appropriate citation (GOM Correlate Software, GOM GmbH. Braunschweig, Germany, 2019) has been made.

Ad. 3. Provide the specs of the glued finger-joints in the timber frames used in the test.

Also, thank you very much for this remark!

The finger-joints were made with accordance to EN 15497:2014: Structural finger jointed solid timber - Performance requirements and minimum production requirements.

The spec has been addressed in the text, in the section 2.1 as: “The timber plates and posts were joined longitudinally using glued finger-joints according to EN 15497:2014”.

Ad. 4. Is there any future works related to this topic?

Yes, these studies will be continued. Further tests in terms of wood-on-wood load perpendicular to the grains are planned.

Reviewer 2 Report

The article shows the results of a very interesting search. The research topic of the manuscript is novel and it fits on the journal purpose.

The reviewer recommends the publication of this paper.

At the same time, the following changes are required to the authors.

  1. The introduction will need to be improved by increasing the bibliography references of similar research studies regarding wooden structures. For example can be suggested the following references: https://doi.org/10.3390/ma11010007
  2. To help the reader, the reviewer suggests entering a table where the results obtained and the value of the compression force of Eurocode 5, are compared.

Author Response

Firstly, the authors are grateful for all valuable comments received from reviewer which allow improving our article!

Our answers to the reviewer’s comments and remarks are given below, in a point-by-point manner.  

  1. The introduction will need to be improved by increasing the bibliography references of similar research studies regarding wooden structures. For example can be suggested the following references: https://doi.org/10.3390/ma11010007

Thank you very much for this recommendation!

The relevant reference articles have been cited. In particular, the reference article https://doi.org/10.3390/ma10070770 was found suitable.

In addition, e.g such works have been cited:Malesza et al. 2019, Branco et al. 2017, Sartori et al. 2013, Madsen et al. 2000, Blass & Görlacher 2004, Leijten 2010, Van der Put 2008, Bodig & Jayne 1982, Dinwoodie 2000, Leijten 2016, Požgaj et al. 1993, Kretschmann 2008, Tabarasa & Chui 2001, Verbist et al. 2020, Kollmann & Côté 1968, Leijten & Jorissen 2010, Kathem et al. 2014, Basta et al. 2011, Stoilov et al. 2019, Speranzini et al. 2017, Johansson et al. 2015.

DOI numbers of some referenced articles are as follows:

https://doi.org/10.3846/jcem.2019.7738

https://doi.org/10.3390/buildings7030063

https://doi.org/10.1016/j.engstruct.2013.08.039

https://doi.org/10.3390/buildings10010014

https://doi.org/10.1515/HF.2011.104

https://doi.org/10.3390/ma10070770

https://doi.org/10.1007/s00107-014-0851-3

  1. To help the reader, the reviewer suggests entering a table where the results obtained and the value of the compression force of Eurocode 5, are compared.

Thank you for your suggestion!

The Table 3 has been added.

Reviewer 3 Report

The authors focus on an interesting problematics, namely "compressive strength perpendicular to the grains of timber frame sill plates from the perspective of their load bearing capacity". The results obtained are in any case beneficial from the perspective of the basis for further research and targeted to a possible change in the document EUROCODE 5.

I would suggest to make few corrections and additions.

Formal shortcomings:

  • failure to comply with the Word document requirements „materials-template“ with regards to the titles of chapters 2 and 3, chapter 4 is not in the text and even so "Conclusions" are marked with number 5,
  • unsuitable name for wood with a moisture content of 10 to 12% (dehydtated),
  • insufficient range of the used literature.

Missing requirements:

  • statistical significance of the obtained results; but it’s all right, this is a requirement that is more suitable for wood of small dimensions, respectively for files under 30 test samples, it is used to verify the statistical significance of the obtained results, through the quantile of Student's distribution (e.g. level 0.95), related variability of measurement (COV) and selected accuracy of measurement (e.g. 5%). However, it is true that Weibull distribution datas seem to be applied more for wood of construction dimensions,
  • the data on the input material of spruce wood in my opinion are not much sufficient, respectively it is only specified by strength class C24. Personally, on the basis of the obtained results (negative conclusions) I would also determine the density and compressive strength perpendicular to the grains according to standardized test procedures applicable for ideal small samples (for construction dimensions, of course, norm EN 408 is normally applied).
  • I would supplement the literature with the standard problematics of strength perpendicular to the grains occurring at least in many basic book publications from the wood science, e.g. Bodig and Jayne 1982, Dinwoodie 2000, Kolmann and Côté 1968, etc.

I consider the used research methods and the introduction to the problematics appropriated and sufficient. I definitely welcome the use of modern research technologies (see e.g. optical system ARAMIS) and relevant discussion of the results.

However, the reproducibility of the results is difficult to estimate, especially due to the high variability of the wood properties (even if in this case it was about a specified strength class). I think it would be appropriate to verify your results also on another type of wood.

Compressive strength perpendicular to the grains on wood is a very specific way of loading, with a three-phase stress-strain diagram typical for metals, i.e. with the yield strength, which is for wood under this strain so-called conventional strength limit. On small samples, e.g. Požgaj et al. (1993) states radial direction approx. 3.4 MPa and in tangential direction 4.0 MPa (higher data in tangential direction is related to the participation of summer wood from the beginning of loading).

Author Response

Firstly, the authors are grateful for all valuable comments received from reviewer which allow improving our article!

Our answers to the reviewer’s comments and remarks are given below, in a point-by-point manner.  

1. Formal shortcomings:

1.1. failure to comply with the Word document requirements „materials-template“ with regards to the titles of chapters 2 and 3, chapter 4 is not in the text and even so "Conclusions" are marked with number 5,

Thank you for this remark!

The titles have been corrected, also wrong chapter 5. Conclusions has been properly marked with No. 4.

1.2. unsuitable name for wood with a moisture content of 10 to 12% (dehydtated),

Thank you!

We absolutely agree, “dry wood” term was used instead.

1.3. insufficient range of the used literature

Thank you for your suggestions!

The appropriate references have been cited.

In particular, e.g such articles have been cited: Malesza et al. 2019, Branco et al. 2017, Sartori et al. 2013, Madsen et al. 2000, Blass & Görlacher 2004, Leijten 2010, Van der Put 2008, Bodig & Jayne 1982, Dinwoodie 2000, Leijten 2016, Požgaj et al. 1993, Kretschmann 2008, Tabarasa & Chui 2001, Verbist et al. 2020, Kollmann & Côté 1968, Leijten & Jorissen 2010, Kathem et al. 2014, Basta et al. 2011, Stoilov et al. 2019, Speranzini et al. 2017, Johansson et al. 2015.

DOI numbers of some referenced articles are as follows:

https://doi.org/10.3846/jcem.2019.7738

https://doi.org/10.3390/buildings7030063

https://doi.org/10.1016/j.engstruct.2013.08.039

https://doi.org/10.3390/buildings10010014

https://doi.org/10.1515/HF.2011.104

https://doi.org/10.3390/ma10070770

https://doi.org/10.1007/s00107-014-0851-3

2. Missing requirements:

2.1. statistical significance of the obtained results; but it’s all right, this is a requirement that is more suitable for wood of small dimensions, respectively for files under 30 test samples, it is used to verify the statistical significance of the obtained results, through the quantile of Student's distribution (e.g. level 0.95), related variability of measurement (COV) and selected accuracy of measurement (e.g. 5%). However, it is true that Weibull distribution datas seem to be applied more for wood of construction dimensions,

Thank you for this comment!

The Weibull distribution was chosen due to natural scale elements. In this case, the kn variance unbiased estimator was equal to 1.73 for number of specimens; in our case: n = 32, which is a little bit more than the requirements towards the Student’s distribution.

2.2. the data on the input material of spruce wood in my opinion are not much sufficient, respectively it is only specified by strength class C24. Personally, on the basis of the obtained results (negative conclusions) I would also determine the density and compressive strength perpendicular to the grains according to standardized test procedures applicable for ideal small samples (for construction dimensions, of course, norm EN 408 is normally applied).

Thank you for this important remark!

Wood elements supplied for testing were certified and marked as C24 class, which defines the scope of mechanical properties of wood - and this is what we followed in these large-scale tests.

In real construction, the grain direction (tangent and radial) relative to the load direction is often random and difficult to define. Due to the high cost of full-scale tests, we plan to carry out further tests on smaller samples. In these tests, we are going to distinguish between tangential and radial grain directions in relation to the load direction. In addition, we plan to test influence of the wood density.

2.3. I would supplement the literature with the standard problematics of strength perpendicular to the grains occurring at least in many basic book publications from the wood science, e.g. Bodig and Jayne 1982, Dinwoodie 2000, Kolmann and Côté 1968, etc.

Thank you for your suggestions!

These books, as well as other references e.g. listed above (in answering p. 1.3) have been supplemented.

2.4. I consider the used research methods and the introduction to the problematics appropriated and sufficient. I definitely welcome the use of modern research technologies (see e.g. optical system ARAMIS) and relevant discussion of the results.

Thank you for your kind evaluation of our research techniques!

2.5. However, the reproducibility of the results is difficult to estimate, especially due to the high variability of the wood properties (even if in this case it was about a specified strength class). I think it would be appropriate to verify your results also on another type of wood.

In the tests, the type of wood was deliberately limited to spruce only, due to its use in construction made by the contractor. Nevertheless, we plan to continue this research. We will also try to publish the next test results.

2.6. Compressive strength perpendicular to the grains on wood is a very specific way of loading, with a three-phase stress-strain diagram typical for metals, i.e. with the yield strength, which is for wood under this strain so-called conventional strength limit. On small samples, e.g. Požgaj et al. (1993) states radial direction approx. 3.4 MPa and in tangential direction 4.0 MPa (higher data in tangential direction is related to the participation of summer wood from the beginning of loading).

Thank you for your comprehensive explanation.

We agree that wood strength varies in radial and tangential directions. We are also aware how it varies in real structure.

This manuscript is a resubmission of an earlier submission. The following is a list of the peer review reports and author responses from that submission.

Round 1

Reviewer 1 Report

General comments

The English deserve some attention. Review by a native speaker is highly recommended. In general, the structure of the paper lacks of rigour. For example, the separation of the text in paragraphs seem to follow no criteria. Some sections are wrongly allocated in the subdivisions. Examples are:

Lines 34-38 in introduction are part of the methodology

Line 53 in the introduction should be in results and discussion

Other general improvements are: uncertainties are missing in some parts, the quality of figures must be improved, and there are some units missing. Reference [1] is cited 13 times in the paper.

But probably the most important is the lack of discussion of the results. The authors limit the results and discussion section to just describe the data without any discussion of the significance of the results obtained (this is most evident, for example, in Figure 9)

Specific comments

Abstract: In line 14, text reads: “the wood moisture was low, which corresponds to the real working conditions...” . If authors claim MC is low, has to be stated with respect what. Also, if MC is same as working conditions, why is it low?                Line 19: the word “dangerous” seems inappropriate

Introduction: See comment above for lines 34-38 and 53

Materials: There is no justification about the specimen geometry chosen.

Line 57 mentioned the timber was joined using glued finger-joints. Which timber? Plate, beams, all? Where are the joints? How many?

Line 66: Symbol error 10-12

Figure 1: units missing

Table 1: not necessary as all the information is repeated in the text.

Line 88: what is the top plate? This was not described or mentioned before

Line 95: time and uncertainties must have the same units (NOT min and sec). Maybe it is better to say the testing time was between 3 and 7 minutes.

Figures 4 and 5: General views are not necessary.

Results: The first paragraph just repeats information. Some information belongs to methodology, not here.

Line 129: Why the process is iterative?

Lines 139-145: In this section there are several parameters that need to be defined (a, b, l, kc, etc). Why the assumption of line 142?

Table 2: The caption does not seem appropriate for a journal paper.

Figure 7: These plots are unacceptable. Authors should find another way to represaent these results.

Figure 8: These could have been cropped for better presentation. The figures on the left side do not provide much information.

Figure 9: error bars missing

Reviewer 2 Report

This paper discusses the compression perpendicular to the grain of timber sills, although this is a useful piece of work if executed properly, there might be a technical flaw that I would suggest the authors to further clarify.

In the drawing I have attached, we can see that the author mount the LVDTs by the inner side of the posts, this can measure disp1, which is not the total deformation of the timber in compression. If the author mount the LVDTs outside the post as shown in the drawing, this will enable authors to measure the total deformation of the timber. Alternatively, as the authors have used Digital Image Correlation (DIC), so in theory they can work out the total deformation. The problem of not using the right displacement is that this will make Load-Displacement curves invalid which will in consequence invalidate the conclusions.

Some other minor comments: 

Line 65-66, the moisture content should be 10-12% not 10÷12%. Since DIC has been used in this paper, the authors could have shown the strain field of the timber in compression this will make the whole paper more interesting.

Round 2

Reviewer 1 Report

The paper is suitable for publication in the present form, but some minor changes are required in Figs 8 and 9

Figs 8a and 9a require an explanation of what it is shown there, besides the points marked. Figs 8b and 9b need a proper Y-axis explicitly saying that the plots show the displacement dx

Author Response

First of all, Authors would like to thank the reviewer for the comments.

Authors corrected the Figures 8 and 9 accordingly: in Figures 8a and 8b, relevant descriptions were added, and in Figures 8b and 9b the Y axis was additionally described.